# Characterization and Mechanism of Gel Deterioration of Egg Yolk Powder during Storage

**DOI:** 10.3390/foods12132477

**Published:** 2023-06-24

**Authors:** Yang Tian, Songyi Lin, Zhijie Bao

**Affiliations:** National Engineering Research Center of Seafood, Liaoning Engineering Research Center of Special Dietary Food, School of Food Science and Technology, Dalian Polytechnic University, Dalian 116034, China; tianyang20200901@163.com (Y.T.); linsongyi730@163.com (S.L.)

**Keywords:** egg yolk powder, accelerated storage test, gel properties, lipid migration, protein oxidation

## Abstract

Egg yolk forms have several health and industrial applications, but their storage characteristics and gel mechanisms have not been thoroughly studied. In order to investigate the relationship between the changes in structure and properties of egg yolk gel and egg yolk powder during storage, in this paper, egg yolk powder was stored at 37 °C for 0, 1, 3, and 6 months in an accelerated storage experiment, and the influence of storage time on the gel properties of egg yolk powder was analyzed. The results showed that the contents of protein carbonylation and sulfhydryl in the yolk decreased gradually with the extension of storage time. Circular dichroism and fluorescence spectra showed that the ordered structure and structural stability of egg yolk proteins decreased gradually. Oxidation led to the formation of intermolecular crosslinking in the egg yolk proteins and oxidized aggregates, resulting in a decrease in surface hydrophobicity, which affected the gel properties of the egg yolk powder after rehydration, resulting in the phenomenon of lipid migration and gel degradation. The results provide a theoretical basis for improving egg yolk powder’s overall quality and storage stability.

## 1. Introduction

Eggs are inexpensive, are a major source of high-quality protein and lipids, and provide many minerals and vitamins [1]. However, fresh eggs are easily spoiled and broken; they are unsuitable for long-distance transportation. In order to meet the food industry’s demand for eggs and egg products, spray-drying technology is often used to remove the moisture in egg liquid and process it into egg powder [2]. Compared with fresh eggs, egg powder can significantly reduce the weight of eggs, has higher biological safety and a shelf life of 1–2 years, and has been widely used in bakery products, mayonnaise, pasta, biscuits, and other food production domains [3]. During the storage process, egg yolk powder (EYP) is inevitably affected by various external environments, such as time, temperature, humidity, oxygen content, and light, which change the quality of the egg yolk powder. From pudding, egg tart, and other processing enterprises, problems are fed back concerning fat precipitation when egg yolk powder is stored for some time after its rehydration; heating to form egg yolk gel also entails a lot of oil precipitation, which affects the taste and flavor of the egg yolk powder itself, and then affects the application prospects of the egg yolk powder.

Egg yolk is a natural multiscale protein–lipid supramolecular assembly [4]. Its microstructure shows many spheroid particles with diameters between 0.8 and 10 μm embedded in the continuous phase [5]. The yolk liquid can be separated into precipitated yolk granules and yolk serous liquid with a centrifuge. The yolk granules are mainly composed of 70% high-density lipoprotein (HDL) micromicelles, 16% vitellophin, and 12% low-density lipoprotein (LDL) [6], while the yolk serous liquid is mainly composed of 85% low-density lipoprotein and 15% vitelloin [7]. LDL is structured as a spherical nanoparticle (17–60 nm) with a phospholipid membrane embedded with proteins surrounding a lipid core composed of triglycerides and cholesterol in a fluid state [8]. HDL is a dimer spherical molecule (7–22 nm) formed by two monomer molecules and presents a spherical micellar structure. At the same time, HDL is coupled with egg yolk hyper phosphoprotein and embedded LDL vesicles to form the main structure of egg yolk granules [9,10].

The gel mechanism of egg yolk liquid follows Flory theory. The formation process is mainly that egg yolk protein changes from a natural to denatured state, then forms soluble aggregates with high relative molecular weight through β folding. Finally, under the action of a disulfide bond, the aggregates gradually thicken to form a gel [11]. However, the difference with egg yolk gel is that the egg yolk liquid is a stable emulsion system. After heating, an emulsion particle gel will be formed, in which egg yolk protein acts as both an emulsifier and a gel matrix [4]. Therefore, the fat precipitation and lipid migration of the gel after the rehydration of yolk powder after storage are bound to be related to the property changes in yolk protein during storage.

Therefore, by accelerating the storage of yolk powder, this paper investigates the physicochemical property changes in yolk protein in yolk powder during storage and attempts to analyze the relationship between these physicochemical property changes and the precipitation of egg yolk gel lipids to provide a theoretical basis for solving the problem of the reduction in processing characteristics of yolk powder during storage.

## 2. Materials and Methods

### 2.1. Samples and Materials

The egg yolk powder used in this study was sourced from Shendi Agricultural Branch Trade Co., Ltd.,Wuhan, China. The protein content of the egg yolk powder was determined to be 31.34 ± 0.58% (*w*/*v*) using the BCA method. The lipid content was 61.22 ± 2.58% (*w*/*v*) as determined by the AOAC-recommended method, and the moisture content was 2.85 ± 0.27% (*w*/*v*). Dithiothreitol (DTT) (CAS: 3483-12-3) and 5,5-dithiobis-(2-nitrobenzoic acid) (DTNB) (CAS:69-78-3) were obtained from Sigma (St. Louis, MO, USA), while all other reagents were of analytical grade and purchased from Solarbio Science & Technology Co., Ltd. (Beijing, China).

### 2.2. Sample Preparation

Accelerated shelf-life testing: The egg yolk powder was packed every 200 g in aluminum foil bags sealed using a vacuum and placed at 37 °C to avoid light for the accelerated test. After 0, 1, 3, and 6 months of storage, the egg yolk powder samples were taken for follow-up analysis.

Egg yolk gel preparation methods: EYP was mixed with 10 mM sodium phosphate buffers (pH 7.4) at a 1:1 ratio to restore the state of the egg yolk liquid. An amount of 20 mL well-stirred egg yolk was stored in a 25 mL beaker, which was sealed with plastic wrap to prevent water evaporation. Then, each beaker was boiled at 100 °C for 5 min to prepare egg yolk gel to explore lipid migration.

### 2.3. Protein Carbonyl Measurements

The protein carbonyl content was determined using a modified version of the method described by Bao et al. [12] EYP samples were dissolved in deionized water at a concentration of 3 mg/mL. Next, 1 mL of the EYP solution was mixed with 3 mL of a 10 mmol/mL solution of 2,4-dinitrophenylhydrazine (DNPH) and incubated at 25 °C for 2 h. Unreacted DNPH was removed using trichloroacetic acid and an ethanol/ethyl acetate solution. The results were expressed as nmol of carbonyl groups per milligram of soluble protein, with a molar extinction coefficient of 22,000 M^−1^ cm^−1^.

### 2.4. Free Sulphydryl Group Measurements

The oxidative changes in proteins during EYP storage were assessed by monitoring free sulphydryl groups (−SH) in proteins using the dithiothreitol colorimetric method, as reported by Chorna et al. [13]. The absorbance was measured at 412 nm. The results were expressed as nmol of −SH per milligram of soluble protein with a molar extinction coefficient of 13,600 M^−1^ cm^−1^.

### 2.5. Low-Field Magnetic Resonance Imaging Analyses

MRI analysis was measured according to Cheng et al. [14] with a little modification. Proton density images were acquired using a multi-spin-echo (MSE) imaging sequence. The specific parameter conditions were as follows: the field of view (FOV) was 70 mm × 70 mm, the slice width was 3.0 mm, read size = 256, phase size = 192, echo time (TE) = 20 ms, and the repeat sampling time (TR) was 2000 ms. Weighted imaging of the samples was obtained through the above parameters, and the images were processed using the software Osirix (Osirix life version 7.0.4, Geneva, Switzerland) for pseudocolor and quantitative processing.

### 2.6. Scanning Electron Microscope Analyses

A small sample piece was obtained from the top part and bottom part of the heated yolk gel after being freeze-dried in the vacuum. Then, the samples were coated with gold on a bronze stub. The microcosmic morphology of the egg yolk gel’s top part and bottom was obtained using an SU8010 (Hitachi, Ltd., Tokyo, Japan) scanning electron microscope in low-vacuum mode with an accelerating voltage of 10 kV, 10 μA and a working distance of 8 mm.

### 2.7. Determination of Spin–Spin Relaxation Time (T2)

According to the method described by Aursand et al. [15], the spin relaxation time (T2) of the sample was measured using a low-field pulsed NMR analyzer (Suzhou Niumag Co., Ltd., Suzhou, China). Test conditions: the instrument was set to CPMG sequence mode to collect attenuation signals from the sample; 90° pulse time was set to 20 μs and 180° pulse time was set to 200 μs; the cumulative number of repeats was 4 times; the waiting time was 3000 Ms; and the number of echoes was 8000. Using the SRIT algorithm of Multiple-Exp Inv Analysis software to invert data, a multiexponential fitting curve was obtained with 1,000,000 iterations.

### 2.8. Differential Scanning Calorimetry (DSC) Analyses

The DSC analysis of EYP was performed using a differential scanning calorimeter (DSC-250, TA Instruments, New Castle, DE, USA) calibrated with indium. Samples of EYP weighing 20.0 mg were placed in disposable aluminum hermetic cells and subjected to heating scans from 20 °C to 220 °C at a rate of 5 °C/min. The melting profiles were analyzed using TA-TRIOS software (TA Instruments, Version 4.4.0, New Castle, DE, USA), and the enthalpy (ΔH) and denaturation temperature (Td) were extracted.

### 2.9. Particle Size Distributions Analyses

EYP was mixed with 10 mM sodium phosphate buffers (pH 7.4) at a 1:1 ratio to restore the state of the egg yolk liquid. Particle size distribution was detected with an acoustic spectrometer (DT 1200, Dispersion Technology Inc., Bedford Hills, NY, USA). For high-viscosity and high-concentration samples, acoustic methods are more reliable and accurate than conventional light-scattering methods (Dukhin) [16]. The instrument manufacturer provided a calibration program to calculate the particle size distribution (PSD) from the attenuation spectra. Details of the instrument can be found elsewhere (Dukhin) [17].

### 2.10. Raman Spectrum Analyses

Raman spectra were measured using a Horiba LabRAM HR Evolution spectrometer (Horiba Jobin Yvon, Palaiseau, France). The collection parameters were as follows: excitation wavelength, 532 nm; laser power, 300 mW; range, 300–1800 cm^−1^; acquisition time, 60 s; and accumulation, 20 times. Background noise was removed and the baseline was adjusted using Labspec Spectrum 6 software (Horiba Jobin Yvon, France).

### 2.11. Secondary Structure Analyses

The Jasco J-1500 spectropolarimeter (Jasco Co., Tokyo, Japan) was used to determine the protein secondary structure. The measurement parameters included a scanning range of 190–260 nm, scanning speed of 50 nm/min, bandwidth of 1.0 nm, CD scale of 200 mdeg/1.0 d OD, and data interval of 0.1 nm. Prior to testing, the oxidized EYHDL samples (1 mg/mL) were dissolved in NaOH solution (pH 8.0).

### 2.12. Tertiary Structure Analysis

The EYP samples were analyzed for intrinsic fluorescence spectroscopy using Hitachi F-2700 fluorescence spectrometers (Hitachi, Japan). EYP solutions were prepared by dissolving 5 mg/mL in deionized water (pH 8.0) and adjusting the pH with 0.5 M NaOH solution. The measurement parameters included an excitation wavelength of 295 nm, a scanning range of 280–460 nm, and a slit width of 5 nm for both excitation and emission.

### 2.13. Statistics Analyses

The statistical package SPSS 20.0 (SPSS Inc., Chicago, IL, USA) was used to subject all data to statistics through one-way ANOVA. Mean values among treatments were compared and significant differences (*p* < 0.05) among treatments were identified using the least-square difference. All data were presented as means ± standard deviations of triplicate determinations.

## 3. Results

### 3.1. Changes in the Functional Group of Stored Egg Yolk Powder

Protein carbonyl content is currently the most commonly used index to characterize protein oxidation. Table 1 shows the changes in the carbonyl content of egg yolk powder during storage. With the extension of storage time, the carbonyl content of the egg yolk powder increased significantly, from 1.66 nmol/mg to 3.22 nmol/mg, indicating that egg yolk protein was oxidized during storage. The degree of protein oxidation deepened with the extension of storage time. Egg yolk powder has a high content of lipids. After spray drying, the chain oxidation reaction of lipids is caused by high temperature, and a large number of lipid free radicals and lipid active oxidation products are produced. These lipid oxidation products have strong biological activity and often cause the oxidative modification of protein structure and the carbonylation of egg yolk protein. Matumoto-Pintro, P. T. [18] found that lipid oxidation occurred in egg yolk powder during storage, and the TBARS value gradually increased with the extension of storage time. When Raitio, R [19] studied the quality of cauliflower soup powder, it was found that the protein in cauliflower soup powder was oxidized during storage, and the carbonyl content gradually increased. The conversion of some amino acid residues to carbonyl derivatives by reactive oxygen species increases carbonyl content [20]. Alternatively, lipid oxidation produces secondary dicarbonyl products, such as malondialdehyde, which combine with myosin to generate protein-bound carbonyls [21].

The content of free sulfhydryl groups plays an important role in the gel structure of egg yolk powder. Sulfhydryl groups are the precursors for forming disulfide bonds, which play an important role in maintaining the tertiary structure of proteins. Therefore, sulfhydryl analysis is an indispensable analytical method to explore the structural and functional changes in proteins during storage [22]. Also shown in Table 1 are the changes in the free and total sulfhydryl contents of egg yolk powder with different storage times. With the extension of storage time, the content of free sulfhydryl and total sulfhydryl decreased (*p* < 0.05). Generally, reducing free sulfhydryl group content generally means protein denaturation or forming aggregates of intermolecular disulfide bonds or oxidation. Oxidation can change the REDOX state of cysteine in the protein and the equilibrium constant of sulfhydryl/disulfide bond interaction, thereby changing the amount and distribution of the sulfhydryl group and disulfide bond in the protein [23]. The content of disulfide bonds in egg yolk powder decreases, and sulfur-containing compounds with non-disulfide bonds are formed, which is not conducive to gel formation [24]. Sulfhydryl groups on the surface of egg yolk powder can be reversibly oxidized to a disulfide bond and sulfenic acid state and irreversibly oxidized to sulfenic acid and a sulfonic acid state. Generally, the decrease in free sulfhydryl group content indicates the oxidative denaturation of egg yolk powder. Studies have shown that reactive oxygen radicals can quickly react with the sulfhydryl group and convert to sulfonic radicals, thus reducing its content [25]. With the extended storage time, the exposed SH group and free SH group in egg yolk powder will rapidly undergo SH oxidation and SH-SS exchange reaction, resulting in decreased total sulfhydryl content. By analyzing the changes in the total sulfhydryl group and free sulfhydryl group content, it was concluded that oxidation would change the structural characteristics of egg yolk protein to a certain extent.

In recent years, Raman spectroscopy has been increasingly used to determine chemical structures [26]. The Raman spectra shown in Figure 1 were obtained after the accelerated storage of egg yolk powder at different times. In the protein Raman spectra, the spectral peak at 1640 cm^−1^ represents the stretching vibration of C=O, reflecting the carbonyl content. As shown in Figure 1, the spectral peak intensity gradually increased with the storage time, representing an increase in protein carbonyl content. The 500–530 cm^−1^ peak is usually associated with the S-S stretching vibration mode of the disulfide bond C-C-S-S-C-C structure [27]. It was found that the spectral peak intensity in this region decreased gradually with the extension of storage time, and the peak intensity of the egg yolk powder stored for 6 months was the lowest, indicating a reduction in disulfide bond content. The 750 cm^−1^ region represents the C-S vibration peak of cysteine residues, and it was found that the intensity of the spectral peak decreased gradually with the extension of storage time, which indicates that the content of cysteine (free sulfhydryl group) decreased gradually [28]. The above results are consistent with the results of the contents of each substance determined in Table 1. The peak intensity of 1157 cm^−1^ represents the tyrosine content, consistent with a mechanism by which lipid free radicals cause protein damage [29]. Hydrophobic interactions involve hydrophobic groups such as aliphatic amino acids. Compared with the control group, the peak strength of the egg yolk powder samples stored for 1–6 months was significantly increased near 1306 cm^−1^ and 1450 cm ^−1^, and the peak strength of the egg yolk powder samples stored for 6 months was more obvious. Among them, the peak near 1306 cm^−1^ represents the bending vibration of C-H, while the peak intensity of 1450 cm^−1^ is related to the bending vibration of CH_2_ and CH_3_ [30]. These indicate that prolonged storage promotes the interaction between aromatic and aliphatic amino acid residues in egg yolk powder and facilitates protein aggregation through hydrophobic interactions.

### 3.2. Denaturation Degree Analysis of Stored Egg Yolk Powder

DSC can provide the temperature and enthalpic changes in the process of protein denaturation, which can effectively reflect the conformational changes and thermal stability of proteins. Proteins initially in the folded (native) conformation have higher enthalpic values; the protein undergoes conformational transformation (unfolding), and the ΔH value will gradually decrease [31]. As shown in Figure 2, the DSC scanning analysis of egg yolk powder samples under different storage periods showed a major endothermic peak of around 134 °C in the map. Because there were a large number of lipoproteins in the egg yolk powder samples, including low-density lipoprotein and high-density lipoprotein, they could not be distinguished well with calorimetry, so there was only a major endothermic peak on the image [32]. In addition, it was obvious that with the extension of storage time, the corresponding enthalpy (ΔH) of egg yolk powder gradually decreased. Combined with the results shown in Table 1, the increase in storage time led to the oxidation of proteins in egg yolk powder, indicating that conformational changes occurred in the egg yolk protein during storage, which also indicated that storage led to the oxidation and aggregation of most proteins in the egg yolk powder. As a result, the existing natural protein content was reduced, and the thermal stability was reduced.

The particle size of the egg yolk liquid sample is a micron, and the particle size change of the egg yolk sample particle aggregate can be measured using a ultrasonic particle size meter, reflecting the effect of storage time on the degree of lipoprotein aggregation in egg yolk [33]. As shown in Figure 3, with the extension of storage time, the particle size of egg yolk liquid gradually increased from 116.27 μm to 3562.62 μm. Combined with the changes in carbonyl and sulfhydryl content, it was found that oxidation aggregation occurred in the egg yolk protein during storage, and with the extension of storage time, the degree of protein oxidation also increased. Covalent crosslinking occurs between protein molecules or subunits [34], and during storage, proteins fold and assemble under van der Waals forces, hydrogen bonds, hydrophobicity, and electrostatic forces, followed by the formation of a larger protein group, increasing in average particle size [35].

### 3.3. Changes in the Secondary/Tertiary Structure of Stored Egg Yolk Powder

In order to investigate the effect of the spatial conformation change in the egg yolk powder on oil precipitation during storage, the secondary structure changes in the egg yolk powder in different storage periods were analyzed. As can be seen in Figure 4a, the yolk powder had negative peaks at 210 and 225 nm, which is consistent with the two negative characteristic shoulder peaks corresponding to the α-helical structure in this region due to the π-π and n-π transitions of the peptide bond [36]. A positive peak at 195 nm indicates that the yolk powder contains a β-folded structure. The intensity of the positive peak at 195 nm and the negative peak at 210 and 225 nm decreased with the storage time.

The secondary structure of proteins can be characterized by the presence of α-helices, β-folds, β-turns, and random coils. The α-helix structure represents a more ordered arrangement of protein molecules, while β-folds, β-turns, and random coils indicate a looser arrangement [37]. The secondary structure content of yolk protein, as shown in Figure 4b, was determined through software analysis. Egg yolk protein contains four types of secondary structures, with β-turn being the most abundant. The content of random coiling decreased gradually with the increasing storage time of the egg yolk powder, indicating that accelerated storage affected the secondary structure of proteins in the egg yolk powder. At the same time, it was also found that the β-Sheet in the secondary structure was reduced compared with the control group, and these phenomena indicate that protein oxidation occurred during the storage period of the egg yolk powder, leading to reduced stability [38].

As shown in Figure 5, the endogenous fluorescence method was used to determine the fluorescence intensity of the egg yolk powder at different storage periods. It can be seen that the endogenous fluorescence intensity of the egg yolk powder showed a trend of first increasing and then decreasing with the extension of storage time compared with the control group. Among them, the yolk powder samples stored for 6 months showed the lowest fluorescence intensity.

The fluorescence spectrum of the protein is mainly associated with tryptophan [39]. Tryptophan, a common amino acid in food proteins, is sensitive to its environment and ideal for molecular studies to determine structure and protein-folding dynamics and protein–protein interactions [40]. Changes in protein fluorescence intensity reflect the degree of oxidation of tryptophan residues and changes in their microenvironment [41]. However, in egg yolk powder, protein oxidation and lipid oxidation will occur during storage, so the analysis of the results in Figure 5 shows that the decrease in peak intensity may be due to the gradual oxidation of tryptophan by the environment with the increase in storage time, which promotes protein–protein or protein–other compound interactions. Thus, tryptophan is buried in the inner core of the newly formed protein aggregates, a conclusion consistent with the results of Guo et al. [42].

### 3.4. Changes in the Water Migration of Egg Yolk Gelation

In order to better characterize the distribution and migration of hydrogen protons in different phases in the food system, the egg yolk powder was gelated at different storage periods, and its magnetic resonance imaging (MRI) was measured. In this sample, a 50 mL syringe was used to prepare the gel, so the longitudinal section of the gel was selected in the MRI detection (Figure 6a), and the pseudocolor image was obtained through later software processing. It can be intuitively seen from Figure 6b that the red color in the middle and lower part of the sample is darker, and the signal intensity in the upper part is weaker. Compared with the control group, the heated egg yolk gel volume increased with the extended storage time, and the proton signal gradually decreased in the MRI images. The proton signal of the egg yolk powder stored for 6 months significantly differed from that of the control group. In magnetic resonance imaging (MRI), protons with lower signal intensity tend to appear blue, while protons with higher signal intensity tend to appear red [43]. As shown in Figure 6b, the closer the sample is to the upper part, the lower the proton signal may be due to the large amount of lipids in the egg yolk, the oxidation of proteins during storage, and the change in lipoprotein structure, which reduces the lipid-binding ability of proteins and makes lipid molecules separate from lipoproteins. This results in a gradual movement of free lipid molecules to the top region of the yolk gel [44]. However, with the extension of storage time, the structural stability of lipoproteins in egg yolk powder decreased, and a large number of free lipid molecules were produced, so the MRI images of egg yolk powder stored for 6 months were significantly different from those of other groups.

Additionally, LF-NMR was utilized to assess the alterations in water and lipid mobility of egg yolk powder during various storage periods. In LF-NMR, two critical parameters that are associated with the physicochemical structure of the sample are relaxation time and peak area. Relaxation times provide insight into the mobility of different water and fat components, while peak areas reflect the content of each water and fat component [45]. Figure 7a shows the T2 multiexponential fitted relaxation curves of the egg yolk powder at different storage periods. Three peaks (T21, T22, and T23) can be observed from this curve. The T21 peak is around 1 ms, which is the fastest relaxation component and belongs to water or lipids closely related to macromolecules [46]. The T22 peak is around 10 ms and reflects water and lipid molecules in the macromolecular network [47]. Peak T23 is around 100 ms and represents the hydrogen proton of lipids in the yolk [48]. By observing the curve in the figure, it can be found that T22 and T23 are continuous in the egg yolk gel. This phenomenon is consistent with the results obtained from the oil studied by Li et al., which contained two continuous characteristic peaks in the 10–100 ms range [49].

Figure 7b shows the peak area plots of the T2 relaxation curves of gel samples prepared from egg yolk powder at different storage periods. As can be seen from the figure, with the extension of storage time, the change degree of A21 is not obvious, A22 shows a trend of first increasing and then decreasing, while A23 shows a trend of gradually increasing. This may be due to the conversion of part of the T22 fraction to T23 as protein molecules interact more closely with water and lipid molecules with prolonged storage time. However, in egg yolk powder with longer storage time, after the gelation of egg yolk induced by heating, the structure of lipoproteins is unstable and easier to be destroyed, thus reducing the lipid-binding ability of proteins, making lipid molecules separate from protein molecules and form protein–water–lipid complexes, making some changes in the structural and functional properties of proteins [44].

### 3.5. Changes in Microstructure of Egg Yolk Gelation

As shown in the figure, the top part and the inner region of the heated egg yolk gel were, respectively, intercepted, and the microstructure of the egg yolk gel sample was characterized with freezing scanning electron microscopy (SEM). It can be seen from Figure 8a that the microstructure of the upper part of the heated egg yolk gel showed an irregular spherical structure. The volume of the irregular spherical structure gradually increased with the storage time. However, the internal structure of the heated egg yolk gel showed a continuous void structure (Figure 8b), and with the extension of the storage time of egg yolk powder, the cavity increased, and the structure became more loose compared with the control group. The spherical structures in Figure 8a are oil droplets and lipoproteins, and the appearance of Figure 8 may be attributed to the heat-induced denaturation of lipoproteins in the egg yolk, releasing free lipids and migrating up and down [50]. It can be inferred that when the egg yolk is heat-treated, the denatured lipoproteins release more lipids, and then the released lipids migrate up and down. They also serve as solvents to extract more lipids [44]. With the increase in storage time, more free lipid molecules are produced in the egg yolk powder, and the increase in free lipids will make the heated egg yolk gel unfold more thoroughly inside, the structure collapse, and the cavity increase. The upper part of the yolk is bound to more lipids and water by hydrogen or hydrophobic interactions to maintain the spherical structure [8]. These results were consistent with the structural results measured using low-field NMR and MRI and confirmed the lipid precipitation problem in egg yolk powder after rehydration for a period of time.

## 4. Discussion

Protein is an important food component and plays various roles in food quality and stability. However, the interaction between lipid oxidation products and proteins can cause structural changes in proteins, affecting their functional properties [51]. Previous studies have reported that egg yolk powder contains many lipids, and spontaneous chain oxidation reactions will occur during storage. The production of lipid oxidation products in egg yolk powder increased with longer storage time [52]. The change in processing and storage conditions of egg yolk powder will lead to a change in the physical and functional characteristics of egg yolk powder [53], which will affect the storage stability of egg yolk powder and its nutritional characteristics [54,55]. This paper focuses on the processing characteristics of egg yolk powder and mainly studies and analyzes the changes in gel characteristics during storage. The essence of gel properties is a change in the protein system that allows the aggregation of partially denatured protein molecules and the binding of aggregates [56]. The results show that during storage, protein oxidation occurs in egg yolk powder, and covalent crosslinking and aggregation occur, which reduce the stability of the protein structure and leads to changes in the spatial conformation of lipoprotein. The protein gel has a loose three-dimensional network structure, and the activity of the lipid molecules embedded in the structure increases. The lipid molecules aggregate together with the driving force generated by heating and the effect of Oswald aging. Due to their low density, they gradually float up, and the final gel state shows the appearance of lipid precipitation and a loose gel structure.

There are many reasons for protein oxidation. As far as the yolk powder itself is concerned, the interaction between lipid oxidation products and protein oxidation is closely related. First, the protein’s secondary structure is mainly maintained by hydrogen bonds formed by carbonyl and amide groups on the backbone. When oxidation occurs, hydrogen bonds and electrostatic interactions that change the ordered structure of the protein occur. At the same time, disulfide bonds and non-covalent bonds generated during oxidation provide the force for protein polymerization. As a result, the protein particle size increases, the ordered structure is transformed into the disordered structure, and the random coil rises [57]. Secondly, the primary oxidation products produced in the process of lipid oxidation will be cleaved to generate free radicals. As potential initiating factors, free radicals cause the protein reaction system to generate free radicals, thus initiating the chain polymerization reaction and leading to protein aggregation [58]. At the same time, secondary oxidation products generated during lipid oxidation can react with the amino groups of protein molecules (Michael addition reaction), resulting in intrachain and interchain crosslinks of polypeptide chains. This interaction mainly acts on nucleophilic amino acid residues, forming disulfide and double tyrosine bonds, which lead to protein crosslinking [59].

The functions of proteins are diverse, and changes in protein structure will inevitably lead to changes in their functional properties, including food texture, water-holding capacity, emulsification, and dispersion. In order to more accurately study the effects of lipid oxidation on protein structure and function, scholars have established different oxidation models to study the effects of lipid oxidation products on proteins. SHEN et al. [60], when studying the effect of oxidation on the gel properties of pork myofibrillar protein and its binding ability with flavor compounds, found that mild oxidation could induce the partial unfolding of pork myofibrillar protein, thereby increasing the salt solubility of myofibrillar protein and improving the hardness of its gel. Bao [12,61] also studied the effects of malondialdehyde and AAPH on the structure and emulsification of high-density lipoprotein in egg yolk, and found that with the increase in oxidation degree, the emulsification characteristics of high-density lipoprotein changed approximately the same as the emulsification characteristics of egg yolk powder after storage. Combined with related studies, Bao also speculated that the unsynchronization might be the complex composition of egg yolk powder. For example, phospholipids are often used as antioxidants [62], and when the imidazole group of His is degraded, it can effectively quench the -OH radical [63]. The Glu-Leu sequence contributes to the O_2_− quenching capacity [64]. Therefore, lipid oxidation in fat-containing foods may be delayed due to the presence of certain proteins and amino acids, making the oxidation model out of sync with the changes in the processing characteristics of actual stored products.

## 5. Conclusions

The gel properties of egg yolk powder during storage were studied and analyzed through an accelerated storage experiment at 37 °C. The results showed that lipid migration of the rehydrated yolk gel appeared after storage. With the extension of storage time, the content of protein carbonyl increased, the content of free sulfhydryl and total sulfhydryl decreased, the secondary structure of the yolk protein was oxidized, the stability of the yolk protein changed from ordered to disordered, and the structural stability became worse. After storage, the oxidized aggregation of lipoproteins in the egg yolk powder resulted in insoluble aggregates, affecting the egg yolk protein’s gel properties. The experimental results provide a powerful scientific basis for further optimizing the processing parameters of yolk powder, reducing the degree of spontaneous reactions during storage, and improving the processing properties of yolk powder.

## Figures and Tables

**Figure 1 foods-12-02477-f001:**
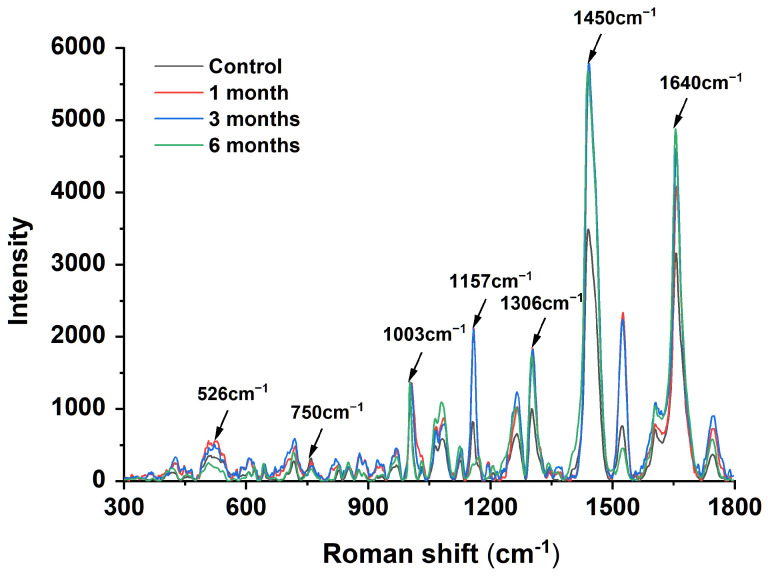
Raman spectroscopy analysis of stored egg yolk powder samples.

**Figure 2 foods-12-02477-f002:**
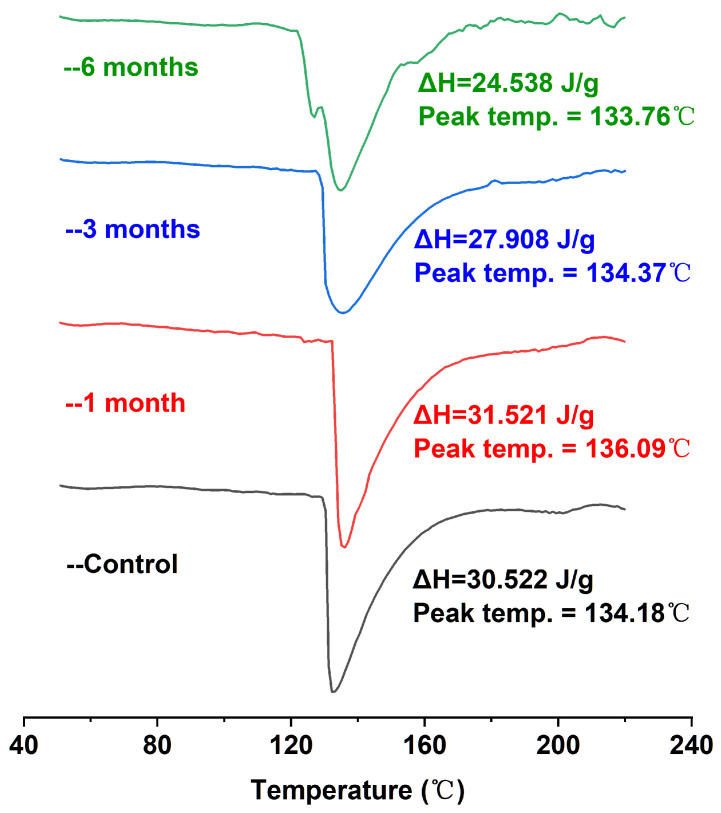
DSC analysis of stored egg yolk powder samples.

**Figure 3 foods-12-02477-f003:**
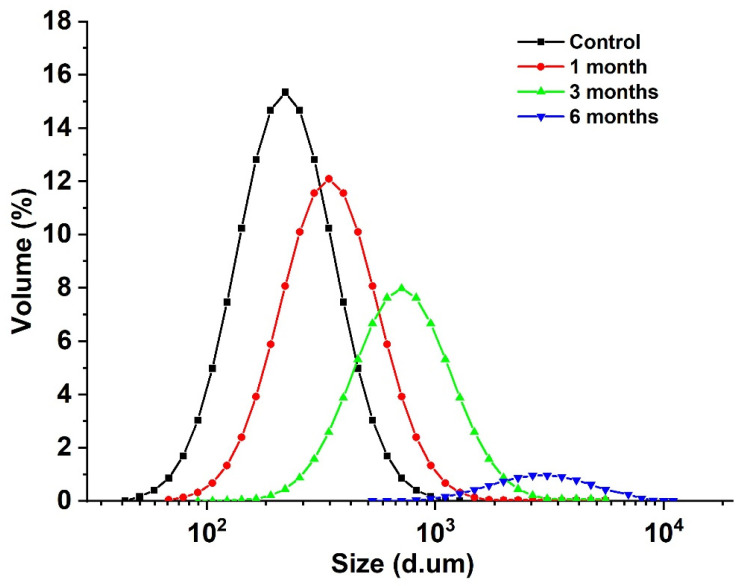
Particle size distribution of stored egg yolk powder samples.

**Figure 4 foods-12-02477-f004:**
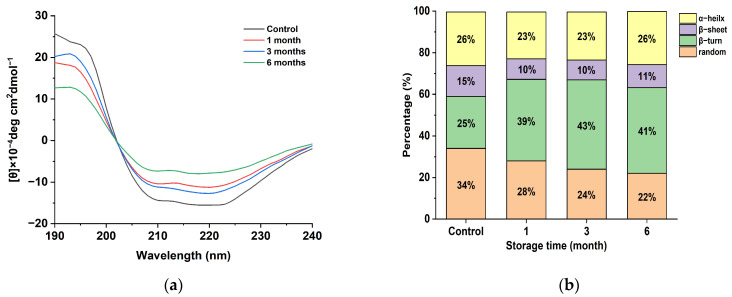
Secondary structure analysis of stored egg yolk powder samples. (**a**) Circular dichroism of oxidized EYHDL samples. (**b**) Secondary structure content of oxidized EYHDL samples.

**Figure 5 foods-12-02477-f005:**
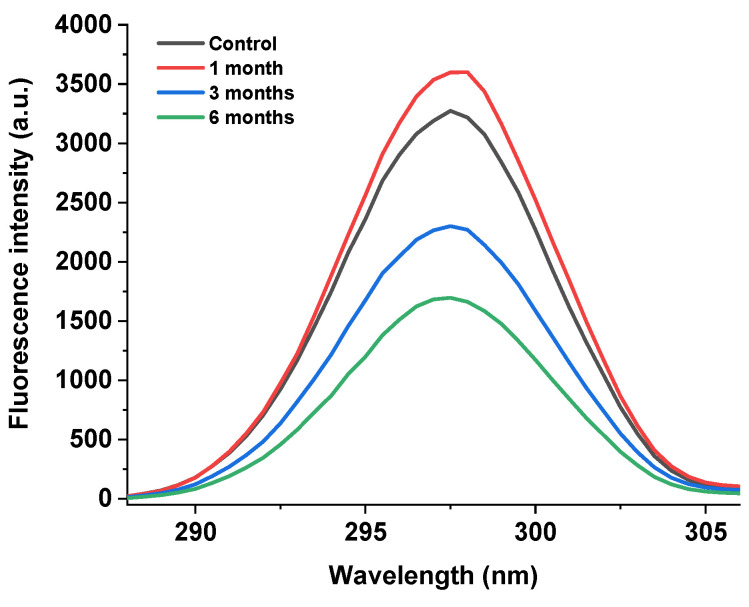
Fluorescence intensity of stored egg yolk powder samples.

**Figure 6 foods-12-02477-f006:**
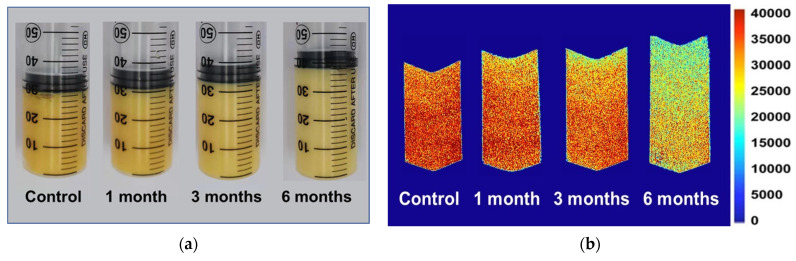
MRI and morphological changes in egg yolk gelation at different stored times. (**a**) MRI changes in egg yolk gelation. (**b**) Images of egg yolk gelation.

**Figure 7 foods-12-02477-f007:**
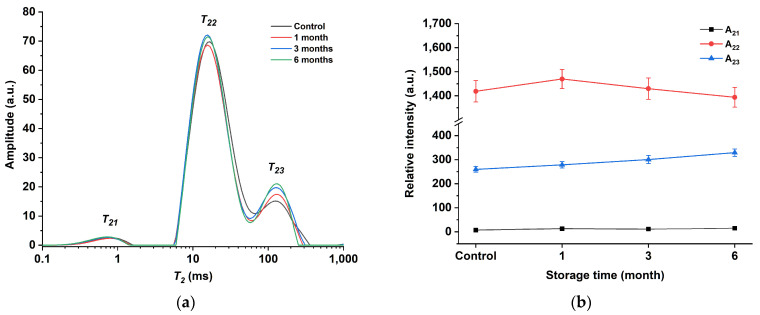
Low-field NMR spectroscopy of egg yolk gelation at different stored times. (**a**) T2 curves of egg yolk gelation. (**b**) T2 peak area of egg yolk gelation.

**Figure 8 foods-12-02477-f008:**
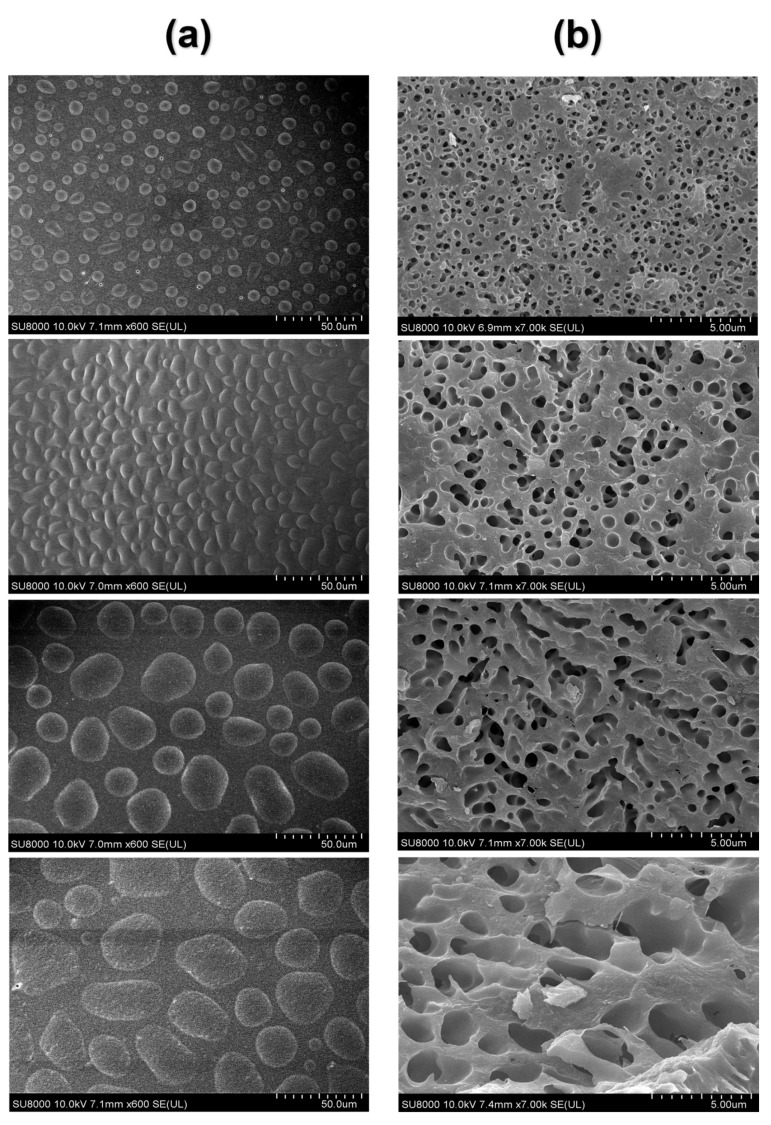
SEM images of egg yolk gelation at different stored times. (**a**) surface SEM images of egg yolk gelation. (**b**) inside SEM images of egg yolk gelation.

**Table 1 foods-12-02477-t001:** Carbonyl group, free sulfhydryl, and total disulfide of stored egg yolk powder samples.

Time (Month)	Carbonyl Group (nmol/mg)	Free Sulfhydryl (nmol/mg)	Total Disulfide (nmol/mg)
0	1.66 ± 0.11 a	2.13 ± 0.12 a	40.4 ± 1.5 a
1	1.86 ± 0.14 b	1.89 ± 0.14 b	38.6 ± 1.4 a
3	2.58 ± 0.13 c	1.51 ± 0.08 c	34.1 ± 1.4 b
6	3.22 ± 0.11 c	1.26 ± 0.11 c	30.1 ± 1.6 c

All values are the means of triplicate determinations. The same letter in the column indicates no significant difference at *p* > 0.05.

## Data Availability

The data presented in this study are available on request from the corresponding author. The data are not publicly available due to them containing information that could compromise research participants’ consent.

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
