# Peer review of "Characterization and Mechanism of Gel Deterioration of Egg Yolk Powder during Storage"

_foods, 2023, doi:10.3390/foods12132477_

Round 1
Reviewer 1 Report
Please make the following revisions to the manuscript:
Change the word "characteristics" to "characterization" in the title.
Lines 11-12: Revise this sentence to avoid starting with "although."
In the abstract, provide a more detailed description of the experimental design.
Include storage time and interval period information in the abstract.
Add a concluding statement to the abstract.
Line 26: Choose either "however" or "because" to avoid confusion.
Lines 50-57: Revise this paragraph for clarity and coherence.
Improve the introduction by focusing more on gel deterioration issues and providing a stronger connection between the provided information.
Include package details and gas transmission rate information in the appropriate section.
Line 154: Correct the typo "nThe" to "The."
Clarify the arrangement of results and discussion sections; consider separating them if necessary.
Explain how storage decreased the sulfhydryl content and why this effect was not severe compared to the total disulfide content.
Figure 1: Add an X-axis label.
Line 223: Correct the chemical formula numbering to subscript and ensure the wavelength unit "cm-1" is in superscript.
Correlate the results in Table 1 with DSC analysis, and explain the fluctuations observed in comparison to the control sample.
Figure 4a: Clarify if the label is the X-axis title or the unit for the X-axis title.
Ensure that the abbreviation "FL" is properly defined in the text.
Arrange SEM images vertically and make them large enough to occupy the entire page for better visibility.
Reevaluate the need for Section 4 if the information has already been discussed in Section 3.
Expand and elaborate on the conclusion, discussing the significance of the study, interpreting the data, and providing recommendations.
English needs extensive revision, many areas in the text are very confusing.
Reviewer 2 Report
The Authors investigated the changes in gelation properties of egg yolk powder during storage. Very interesting study. There is no in the literature a similar paper. English language must be corrected by a native speaker.
Some important papers are missing, which should be used in the Introduction and/or in the Discussion.
Li et al. (2006) investigated Cholesterol Oxidation in Egg Yolk Powder During Storage and Heating as Affected by Dietary Oils and Tocopherol. Journal of Food Science. 61. 721 - 725. 10.1111/j.1365-2621.1996.tb12189.x.
The effects of feeding flax, sunflower, palm and fish oils, with and without tocopherols, to laying hens was investigated on oxidative stability of cholesterol in egg yolk powders. Storage of spray-dried egg yolk powders at room temperature resulted in loss (p <0.05) of tocopherols. The initial levels of spray-dried yolk cholesterol oxides were 7–10 ppm and increased (p <0.05) with storage. The cholesterol oxide contents in the egg yolk powders were: fish > flax > sunflower > palm. Heating egg yolk powder increased (p <0.05) the amounts of total cholesterol oxides.
Rannou et al. (2015) looked at the effect of spray-drying and storage conditions on the physical and functional properties of standard and n−3 enriched egg yolk powders. Journal of Food Engineering, 154, 58-68. https://doi.org/10.1016/j.jfoodeng.2014.11.002.
This study aimed to evaluate the effect of the processing and storage conditions on the physical and functional properties of egg yolk (EY) powders. The spray-drying temperature (160 °C vs. 180 °C), storage temperature (15 °C vs. 30 °C) and time (1, 2, 4 and 8 months) and n−3 enrichment through hen diet were investigated. The spray drying temperature and storage conditions modified the water content, water activity and the particle size distribution of EY powders. Flowability of the powders and the emulsifying properties were not significantly affected from an industrial point of view. On the opposite, the viscosity increased with the spray-drying temperature as well as the temperature and time of storage in rehydrated powders. Powders prepared with n−3 enriched egg yolks exhibited lower melting peaks temperatures, a marked yellow colour and higher fluidity of the solutions, but the overall properties remained unchanged.
Rao et al. (2013) investigated storage Stability of a Commercial Hen Egg Yolk Powder in Dry and Intermediate-Moisture Food Matrices. Journal of Agricultural and Food Chemistry, 61, 8676-8686. DOI: 10.1021/jf402631y
Quality loss in intermediate-moisture foods (IMF) such as high-protein nutrition bars (HPNB) in the form of hardening, nonenzymatic browning, and free amino group loss is a general concern for the manufacturers. To measure the extent of quality loss over time in terms of these negative attributes, through changing the ratio by weight between two commercial spray-dried hen egg powders, egg white (DEW) and egg yolk (DEY), the storage stability of 10 IMF systems (water activity (aw) ∼ 0.6) containing 5% glycerol, 10% shortening, 35% protein, and 50% sweetener (either maltitol or 50% high-fructose corn syrup/50% corn syrup (HFCS/CS)) were studied. Additionally, the storage stability of the DEY powder itself was investigated. Overall, during storage at different temperatures (23, 35, and 45 °C), the storage stability of DEY in dry and IMF matrices was mainly controlled by the coaction of three chemical reactions (disulfide bond interaction, Maillard reaction, and lipid oxidation). The results showed that by replacing 25% of DEW in an IMF model system with DEY, the rate of bar hardening was significantly lower than that of the models with only DEW at all temperatures due to the softening effect of the fat in DEY. Furthermore, the use of maltitol instead of HFCS/CS in all bar systems not only resulted in decreased hardness but also drastically decreased the change in the total color difference (ΔE*). Interestingly, there was no significant loss of free amino groups in the maltitol systems at any DEW/DEY ratio.
Wenzel et al. (2011) investigated influences of storage time and temperature on the xanthophyll content of freeze-dried egg yolk. Food Chemistry, 124, 1343-1348. https://doi.org/10.1016/j.foodchem.2010.07.085.
The influences of storage time, temperature (−18 or +20 °C, both in the dark), and prior pasteurisation on the xanthophyll content of freeze-dried egg yolk were investigated. After six months of storage, the synthetic xanthophylls all-E-canthaxanthin and β-apo-8-carotenoic acid ethyl ester showed considerably higher stability (with losses of 19–34%) than did the natural pigments all-E-lutein and all-E-zeaxanthin (losses of 59–69%). At all stages of storage, the xanthophyll contents of unpasteurised and previously pasteurised samples did not differ significantly, and no obvious influence of storage temperature was observed. With respect to xanthophyll content, the results suggest that there is no necessity for low-temperature storage of freeze-dried egg yolk.
Very important paper on heat-induced gelation of yolk is missing.
Cordobés et al. (2004) investigated rheology and microstructure of heat-induced egg yolk gels. Rheol Acta 43, 184–195. https://doi.org/10.1007/s00397-003-0338-3.
The evolution of native egg yolk undergoing a thermal-induced sol-gel transition was studied by using temperature controlled small amplitude oscillatory shear measurements. The critical gel point was determined according to Winter’s criterion: 1) from the measurements of storage and loss moduli as a function of heating time at different frequencies, and 2) from the exponents of the power law mechanical spectra obtained after cure experiments performed up to a maximum temperature (60–90 °C) followed by a sudden decrease in temperature up to 20 °C. Differential Scanning Calorimetry (DSC) was performed in order to investigate thermal transitions in egg yolk. Microstructure of gels was evaluated by Transmission and Scanning Electron Microscopy. The results obtained were discussed in terms of the processes involved in protein gelation: change in the protein system, aggregation of partially denaturated protein molecules and association of aggregates. As a result, an elastic gel network was always obtained. The influence of frequency, heating rate, solids concentration and maximum temperature of processing, was analysed. Most of the transformations found during thermal processing were found to be basically irreversible, even at the sol state and gel point. However, some reversible phenomena were detected during constant temperature processing depending on the maximum temperature performed.
Inconsistent form of literature positions list. Must correct according to the journal requirements.
English language must be corrected by a native speaker.
Reviewer 3 Report
The work presents interesting results on the changes in biomolecules during the storage of egg yolks.
In materials and methods it should be indicated which methodology was used to determine the protein content. Some method such as Bradford, BCA, Kjeldahl, Dumas was used. An AOAC recommended protocol was followed… lipid method...
The work in general is very well written and has a very good presentation of the text, tables and figures, but the results section is not dedicated exclusively to showing the results. For example, in the first section “3.1 Changes in the functional group of stored egg yolk powder”. The data is quoted lightly and the text enters into a discussion of the assumptions underlying the data. But in no case are the data expressed for one month to six months. The reason for the differences between these data is not explained. There is a text that can perfectly be part of the discussion. This results section should be rewritten to fit the Results format. In the work, lipid peroxidation during egg storage is discussed a lot, but TBARS was not evaluated, nor was SDS-PAGE electrophoresis performed to visualize denatured proteins, different money, trimers.
The bibliographical references section should be reviewed in depth, many of the bibliographical references have different formats. For example, the year is listed in different places and the names of the journals are sometimes abbreviated and sometimes not. Please read the rules for authors
Round 2
Reviewer 1 Report
Authors have revised well enough, at present this paper can be accepted for publications.
N/A
Reviewer 3 Report
Thanks to the authors for accepting comments. I have no more comments.
Best regards